# A Standard Operating Procedure for Dual-Task Training to Improve Physical and Cognitive Function in Older Adults: A Scoping Review

**DOI:** 10.3390/brainsci15080785

**Published:** 2025-07-23

**Authors:** Luca Petrigna, Alessandra Amato, Alessandro Castorina, Giuseppe Musumeci

**Affiliations:** 1Department of Biomedical and Biotechnological Sciences, Section of Anatomy, Histology and Movement Science, School of Medicine, University of Catania, Via S. Sofia 97, 95123 Catania, Italy; alessandra.amato@unict.it (A.A.); g.musumeci@unict.it (G.M.); 2Laboratory of Cellular and Molecular Neuroscience (LCMN), School of Life Sciences, Faculty of Science, University of Technology Sydney, P.O. Box 123, Broadway, Sydney, NSW 2007, Australia; alessandro.castorina@uts.edu.au; 3Research Center on Motor Activities (CRAM), University of Catania, Via S. Sofia n 97, 95123 Catania, Italy

**Keywords:** DT, SOP, protocol, elderly, scoping review

## Abstract

Background/Objectives: Dual task (DT) training consists of practicing exercises while simultaneously performing a concurrent motor or cognitive task. This training modality seems to have beneficial effects on both domains. Various forms of DT training have been implemented for older adults in recent years, but no official guidelines currently exist. This review sought to analyze the studies published on this topic in the last ten years and provide a standard operating procedure (SOP) for healthy older adults in this context. Methods: The review collected articles from PubMed, Web of Science, and Scopus, adopting a designated set of keywords. Selected manuscripts and relevant information were selected, extrapolated, including information related to the training frequency, intensity, time, and type, and secondary tasks adopted. The secondary tasks were grouped according to previously published studies, and the SOP was created based on the frequency of the parameters collected from the included articles. Results: A total of 44 studies were included in the review. Based on the results, the SOP recommends postural balance or resistance training as primary tasks, combined with a mental tracking task as a secondary component. Two 60-min sessions per week for at least 12 weeks are required to achieve measurable results. Conclusions: Despite heterogeneity in the literature reviewed, the findings support the proposal of a SOP to guide future research on DT training in healthy older adults. Given its feasibility and positive effects on both motor and cognitive functions, this type of training can also be implemented in everyday settings.

## 1. Introduction

The dual task (DT) concept, introduced at the beginning of the 21st century, consists of performing a primary and a secondary task simultaneously [1]. The tasks, either primary or secondary, can be motor or cognitive [2]. The aim is to shift attention between the two tasks, and this can be modulated by controlling the difficulty of either the primary or secondary task [3]. The DT concept aims to create a more challenging condition. Indeed, a reduction in postural control (in static condition) has been observed in older adults during DT situations [4]. This concept has also been applied in dynamic situations, such as walking, and in falls among older adults [5]. In the literature, different secondary tasks exist, such as manual tasks, reaction time tasks, discrimination and decision-making tasks, mental tracking tasks, verbal fluency tasks, and working memory tasks [2,6], and they are widely adopted during testing and training. Two recent reviews helped the community to better standardize secondary tasks during static [2] and dynamic conditions [6], providing useful guidelines for the evaluation.

Dual task testing aims to reveal individual risks through the concurrent combination of a primary task (such as a postural balance test or a walking test) and a secondary task (such as a manual or a cognitive task). Examples include studies on postural balance [7] and gait ability [8], which are also adopted to predict possible future falls in healthy [9], and pathological populations [10]. This testing methodology is also adopted to assess cognitive performance [11], influence manual dexterity [12], and evaluate mobility [13,14]. Another example of DT testing is its use as a predictive tool for cognitive impairment [15,16]. Considering the various domains in which the DT concept is applied, it also shows strong potential as a training tool. Consequently, greater attention should be given to DT training.

Training in a DT context means that traditional training can be combined with cognitive training or with a concurrent motor task, or that two cognitive tasks can be combined [3]. This training methodology takes different forms and can be adapted to the participant’s level and context [11]. The advantage of DT training lies in its benefits for both motor and cognitive functions. Indeed, this methodology significantly engages with the left prefrontal cortex and the parietal area [17]. In young adults, DT situations have been shown to increase activity in the prefrontal cortex [18]. This brain region is considered as an executive hub, and is strongly associated with executive function [19]. Consequently, DT training improves executive function more effectively than single cognitive training [20,21]. Dual task training, proposed for healthy older adults, has a positive effect on postural balance [22,23] and also reduces the risk of falls [3]. It is also used for walking speed training [24]. Furthermore, it has shown positive results on cognition [11]. Cognitive and physical function can be improved with DT training not only in heathy individuals, but also in people with dementia or mild cognitive impairment [15,16], or in people with Parkinson’s disease [25], neurological impairment [26], in people suffering from chronic stroke [23], and in those with multiple sclerosis and Alzheimer’s disease [27]. Generally, it seems that this training methodology works in people with neurologic disorders to improve gait, postural balance, and cognition [28].

The American College of Sports Medicine (ACSM) periodically provides updated and clear guidelines for exercise testing and prescription [29]. Specific indications for training frequency, intensity, time, type, volume, and progression (FITT-VP) are provided for different training methodologies across various populations, both healthy and clinical [29]. Despite the DT concept with different testing and training protocols being proposed at the beginning of the 21st century, clear and standardized guidelines have not yet been created. Unfortunately, DT training includes diverse and inconsistent protocols, limiting comparisons between studies [28]. Therefore, a standardized DT training protocol does not exist [15], and there is considerable heterogeneity in this field [26]. The research gap that still exists in this training methodology lies in the variety of approaches, and at times, the impracticality of applying them in everyday contexts [11]. One review highlighted that gaps still exist on the optimal dose and the best approach to adopt [25], and there is also a lack of reporting on key implementation details [11]. Furthermore, it would be valuable to study the most effective training protocols tailored for specific diagnostic groups [28], but this is only possible if a common starting point is established. Standard operating procedures (SOPs) are present in the literature and are defined as detailed documents outlining each step of a procedure [30,31]. They are also used in the field of sports science, providing guidelines and ensuring that procedures are replicable and comparable [32]. Because it is essential to design feasible interventions and achieve positive outcomes, the objective of the present study was to investigate DT training in healthy older adults aimed at improving physical and cognitive function and to propose a SOP for DT training in this population.

## 2. Materials and Methods

The Preferred Reporting Items for Systematic Reviews and Meta-Analyses (PRISMA-ScR) checklist and explanation for scoping reviews [33] was followed. For more details, the PRISMA checklist is included in the Appendix A. The work has not been registered on a specific database, but the protocol was previously written and followed step by step.

### 2.1. Eligibility Criteria

The eligibility criteria of this review were, as suggested by the PRISMA checklist, for a Population, Intervention, Comparison, Outcomes, and Study design (PICO-S). The population investigated was composed of healthy older adults (65 years and above). Studies were excluded if any kind of physical (i.e., muscle or bone injuries or problems, inflammatory joint conditions, pathologies) or cognitive (i.e., dementia, cognitive deterioration) impairment had been investigated. Studies were also excluded if a specific population, such as people with Parkinson’s disease, multiple sclerosis, fibromyalgia, and similar conditions, had been investigated.

The intervention had to be characterized by training based on the DT principle. The training had to include a primary task performed with another task, called a secondary task [1]. The primary task had to be motor training, while the secondary task could be a motor or a cognitive task. Only interventions that did not require specific technologies (i.e., televisions, computers, tablets, consoles, or three-dimensional environments) were considered to make our findings replicable in any context with zero costs. Studies that adopted virtual reality, specific games and consoles, or specific settings were excluded. Specific dance programs not based on dual-tasking were also excluded. Acute interventions were excluded as we aimed to focus attention on something created to obtain effects in the long-term period.

Because a meta-analysis was not performed, our types of comparison and outcomes were all accepted. Our main goal was centered on the protocol adopted in the studies.

Studies included had to present a study design that could be cross-sectional, longitudinal, correlational (both randomized and non-randomized controlled trials, and quasi-randomized studies), or intervention studies. The included studies had to be original, peer-reviewed, published in international journals, and had to be written in English. Pre-print and protocol studies were excluded. Other kinds of study designs or those written in a different language were excluded. Considering the trend of published articles on PubMed, the studies were included if published from 2015 (in the last ten years).

### 2.2. Data Collection

Studies were collected from three different electronic databases (PubMed, Web of Science, and Scopus). All articles published until 25 March 2025 were included. Four groups of keywords were adopted and matched with the Boolean operators AND/OR to create the following string:Group 1: dual task, double task;Group 2: training, protocol, practice, exercise;Group 3: cognitive function, cognition;Group 4: older adults, elderly.

The string generated was composed of: (dual task OR double task) AND (training OR protocol OR practice OR exercise) AND (cognitive function OR cognition) AND (older adults OR elderly).

The string was adopted in the three databases in the current form. No filters were used.

### 2.3. Study Record

The software EndNote (EndNote version X8; Thompson Reuters, New York, NY, USA) was adopted to manage the studies. In the first step, duplicates were searched and eliminated. In the second step, two investigators independently performed the study selection by applying the eligibility criteria. The title, abstract, and full-length articles were screened in this order. The studies were included if both investigators agreed, otherwise, the coordinator made the final decision. The investigators were not blinded to the information related to the authors or associated institutions.

When all of the articles had been selected, data regarding the sample size and age of the population, intervention duration, frequency, intensity, time, and type, any adverse events, adherence and dropout rates, single task, and secondary tasks were collected. The information on the secondary task proposed was collected according to the macro-categories presented in previously published articles [2,6]. Furthermore, information about the objective and the DT efficacy was also detected. The SOP was created based on the frequency of the parameters considered (frequency, intensity, time, type, primary and secondary task typology) in the included studies. Data were presented through tables and discussed narratively.

## 3. Results

A total of 2915 articles (PubMed, 1624; Web of Science, 774; and Scopus, 517) were identified. After duplicate removal, 2098 studies were screened against the eligibility criteria. After title selection, 376 abstracts were screened, and a final number of 211 articles remained for the full-text review. After applying the full-text eligibility criteria, a final number of 44 original articles were included in the review. The selection process is presented in Figure 1.

### 3.1. Characteristics of the Studies

A total of 2568 participants were included in the studies, with a sample size ranging from 11 [34] to 158 [35]. The mean age ranged from 62 [36] to 85 years [37,38,39,40]. Two studies [41,42] did not report the participants’ ages but only general demographic information. Two studies reported an age range [37,43]. Overall, the adherence to the training was above 70%, with only one article reporting an adherence below this percentage. The dropout rate was also low, with data showing around 10% of dropouts. Only one article reported a high dropout rate of 50%. Information related to the studies included is presented in Table 1.

### 3.2. Description of the Intervention

The protocol duration ranged from three training sessions [43] to 26 weeks [77]. Most of the studies opted for a 12-week intervention. Some studies presented the data in terms of sessions: 12 [35], 16 [52], and 24 sessions were reported [85]. Four studies did not report the length of the intervention, but it appeared to be long-term. Most of the studies proposed two or three sessions per week; one study did not report this information, and one study adopted a previously published protocol [66]. A session duration of 60 min was mostly adopted. Ranges such as 54–60 and 60–75/80 min were also reported in the included studies.

Most of the studies adopted progressive intensity. Seven studies clearly reported how the intensity was monitored, two adopted the percentages (60–70% and 70–85%) of maximal heart rate, and three adopted the rate of perceived effort. Two studies adopted the 1 repetition maximal. Six studies did not provide information about the intensity or type of intervention. More details are presented in Table 2.

To structure the training session, “warm-up, conditioning, and cool-down” phases were adopted in 18 studies. The detailed length of each phase can be found in Table 1 because most of the studies adopted different protocols. Only the warm-up or the cool-down phases were described in one study. Interesting, amusing, and social tasks were adopted in the cool-down by Blasco-Lafarga and colleagues [65], and beginning and finishing rituals were included in the study of Rosado and colleagues [97]. Fifteen minutes of mental gymnastics (finger motions) during the warm-up was adopted by Yokoyama and colleagues [108]. A useful addition was the 10-min communication time proposed by Sok and colleagues [104]. Regarding the content of the programs, some studies included activities of daily living or similar patterns within resistance, endurance, postural balance, or mobility training [67,71,85,106]. Specific and previously published protocols were adopted in 20 studies, most of which were used once [38,46,50,51,54,55,56,57,58,59,60,61,62,63,64,66,72,74,76,78,82,84,95,96,99,101,102,103,105,107]. The protocols of Halvarsson (2015) and Wollesen and colleagues [63,70] were each adopted in two studies [69,80] and the protocols of Arrieta and colleagues and Rodriguez-Larrad and colleagues [95,96] twice [38,100]. The ACSM guidelines were cited in four studies, but different citations were adopted [48,54,68]. The American Geriatrics Society Panel on Exercise and Osteoarthritis [49] guidelines were adopted in one study.

The most studied aspects were elements related to cognitive function. Postural balance, walking or gait characteristics, and physical fitness were also investigated multiple times. More details are provided in Table 3.

### 3.3. Characteristics of the Dual-Task Training

The primary tasks, secondary task interventions, the outcomes investigated, and their effects are presented in Table 4. The effects of the interventions were mostly positive, with two studies reporting a small effect, five studies reporting no differences with DT training, and the study of Rezola-Pardo and colleagues [39] showing a negative trend. Sequential and simultaneous cognitive and physical training was adopted by Chuang and colleagues [35].

The frequency of the primary and secondary tasks is presented in Table 5. In the included studies, motor tasks were mainly proposed as a primary tasks such as postural balance, resistance training exercises, gait and walking exercises, aerobic training, stepping, mobility exercise, and daily life activities. Less adopted training methodologies are presented in Table 5. Previously published protocols [63,72,82,95,96,99,101,102,103,105,111,122] were adopted in eight studies.

The most adopted secondary tasks were mental tracking tasks. Among them, mathematical calculations, counting backward, or solving anagrams were widely adopted. These tasks were closely followed by working memory tasks such as short-term memory (i.e., evoking names, remembering words and how they are listed, items, shapes, colors, or numbers previously said, counting the days of the week), long-term memory (i.e., completing proverbs or traditional songs), and semantic memory. Verbal fluency tasks were also widely adopted including reading and talking, reverse spelling, and singing songs. Less frequently adopted tasks were reaction time tasks (e.g., simple and choice, using visual and auditory stimuli) and discrimination-decision making tasks (e.g., the ability to inhibit automated responses, problem-solving, task prioritization, and task switching). Ten studies adopted other types of tasks, such as tooth brushing, as in the study by Granacher and colleagues [75]. The Shiritori, a Japanese word chain game in which one player has to say a word starting with the last character of the word given by the previous player, was adopted by Yokoyama and colleagues [108]. Talking was adopted by Anandh and colleagues [43] and a visuospatial skills task by Chuang and colleagues [35], while visual tasks were used by de Maio Nascimento and colleagues [71]. Ansai and colleagues [47] adopted cards of different colors with a description of different activities, requiring the participants to say and execute the action based only on observing the card color. Singing unknown songs, memorizing, and learning the lyrics of songs played in previous sessions was adopted by Nieto-Guisado and colleagues [90]. The emotional prosody task was adopted by Jardim and colleagues [85]. No details were provided in the study by Ho and colleagues [81]. Examples of motor tasks adopted include holding half-filled glasses with both hands, carrying and/or manipulating objects, bimanual tapping, throwing or holding a ball or a bag, balancing a cup on the palm, and tossing a ball from one hand to the other hand. Seventeen studies adopted previously published protocols. The protocol by Halvarsson and colleagues [70] was adopted in two studies [79,80] as was the protocol by Rezola-Pardo and colleagues [103], which was also adopted by two studies [38,39]. Detailed information about the number of studies for each motor and cognitive task is provided in Table 4.

## 4. Discussion

The literature review highlighted heterogeneity and uncertainty in the protocol designs. Despite this, from the protocols of the included studies, a SOP as proposed for healthy older adults, as presented in Table 6, and is discussed in-depth below.

The ACSM guidelines for older adults were followed by different authors, and the primary task proposed in the DT training could follow the latest ACSM guidelines suggested [29] for the reference population. It seems that for the length of the intervention, if the participants were regular and attended the weekly sessions, 12 weeks or more were necessary to obtain positive results [27]. Our SOP, in line with the literature [123], suggests a progression in difficulty as well as task specificity to obtain better outcomes on cognitive functions. The most adopted primary task in the included studies was postural balance, both static and dynamic, such as gait exercises. In line with a recent article [124], postural balance training involves modifying the body position with movement and gesture. It should also change the nature of the ground surface or environment, or use sports equipment [124]. The optimal postural balance training should include various postural tasks that people encounter in physical practice [124]. Reviews [8,23] that examined DT during gait and balance training confirmed more positive outcomes than single task training. This supports the use and feasibility of postural balance training integrated with the DT concept. Consequently, as a practical guide, we strongly suggest including this component to prevent falls or to improve activities of daily living. Among the included studies, resistance training was also widely adopted. When resistance training is proposed, trainers or researchers should consider using machine-based strength training devices. According to the literature [125], this makes it easier to control the volume and intensity (to prevent cognitive components from interfering with strength training). Furthermore, resistance training should also include power training [125]. An example could be combining this training modality with a cognitive task such as a rapid reaction task [125]. From a practical point of view, resistance training must be properly monitored, and the participants provided with adequate instructions and technique to ensure safety [126]. In terms of dosage, 5–8 repetitions with 2–3 sets at 50–80% of 1 RM should be proposed. The trainer should offer 1–2 exercises for the major muscle groups, 2–3 times a week. These guidelines were drawn from both the included studies and the literature [29,126]. If well-planned, resistance training and power training should be included as they have positive effects on muscle mass, strength, and the risk of falls [127]. Additional benefits include musculoskeletal health, psychological well-being, sleep quality, and the management of chronic conditions [128]. Less adopted primary tasks in the included articles were aerobic exercises, mobility, and daily life activity-based exercises. From a practical point of view, high-intensity interval training or moderate-intensity continuous training should be proposed. These approaches yield positive outcomes on cardio-respiratory function, fat distribution, and cognitive function [129]. According to the literature, high-intensity interval training not only benefits the cardiometabolic system, but also improves quality of life [130], while mobility training enhances functional mobility [131]. Activities of daily life, when used as exercise, facilitate the transfer of training effects into daily living [24].

The most frequently adopted secondary tasks in the included studies were mental tracking tasks. These included counting backward, arithmetic tasks, and sequence tasks, three widely adopted categories [2]. From a practical standpoint, counting backward is easy to administer and modulate in terms of difficulty. Counting backward by one is the easiest, followed by three, with seven being the most challenging [2]. Arithmetic tasks (e.g., mathematical problems [2]) can be presented in a game-like format, as can sequence tasks. Mental tracking tasks significantly impact the motor system. A study on walking in young and older adults showed that these tasks, when proposed during walking, shortened stride length, increased stride time, and raised the stride length and variability [132]. This suggests that simple, easy-to-propose tasks can be used to create challenging practice conditions in everyday practice. Working memory tasks were also widely adopted. Common types included n-back tasks, backward digit recall, the Brooks’ spatial/nonspatial memory task, and digit or word memory tasks (e.g., memorizing the greatest number of numbers or words) [2]. According to the literature [133], a strong link exists between physical activity and working memory involving the capacity and efficiency of attentional resources. Working memory training improves memory performance, especially visual working memory [134], which may enhance overall cognitive functioning [135]. Verbal fluency tasks were the third most frequently adopted in the included studies. Examples included sentence completion and repeating numbers aloud [2]. A study on people with Alzheimer’s disease showed that walking speed decreased during a verbal fluency task, [136], which suggests that verbalization increases cognitive–motor interference [23], raising the attentional demand during post-lexical processing [137]. Overall, verbal fluency, discrimination and decision-making, and reaction time tasks appear less challenging than mental tracking and working memory tasks when combined with postural control [138]. The impact of cognitive tasks depends on the attentional resources each task requires, and limitations in central capacity to allocate an appropriate level of attention can influence performance [139].

Motor/manual secondary tasks were also adopted several times such as holding a glass of water or an object, finger tapping, and similar activities [2]. One study found that fallers were less accurate than non-fallers when placing an object during a postural stability task [140]. Unlike cognitive-motor combinations (which use different neuronal networks), motor-motor combinations use the same resources. According to the cross-talk model, similar domain tasks interfere with one another [141]. The bottleneck principle also suggests that two tasks relying on the same neural processors must be performed sequentially to avoid performance interference [142].

It is important that trainers modulate the intensity of the secondary task in both motor-cognitive and motor-motor combinations. According to the literature [24], task prioritization should be varied between primary and secondary tasks. When planning DT training, the personalization and adaptation of both primary and secondary tasks is essential. Training must be challenging but not discouraging, as motivational aspects are crucial to maintain adherence [125].

In terms of outcomes, DT training has positive effects on cognitive and executive function. The literature shows that DT activity leads to structural brain changes such as in the gray (visual motion complex area) and white matter volume [143]. Even in older adults, cognitive and exercise training can induce cognitive plasticity [144]. Moreover, older adults demonstrate the ability to increase neural resource allocation [145].

From the findings of this review, DT training also improves static and dynamic postural balance. The literature shows that adding a motor task to a balance task triggers brain network reorganization, increasing interregional connectivity and central resource expansion [146]. Given that DT relies on attentional resource allocation during postural control [147,148], DT postural training offers a high cognitive challenge. In the included studies, improvements in general physical fitness, falls prevention, mobility, and quality of life were observed. Another review confirmed that more than two sessions per week, each over 30 min, improved global cognition, executive and physical functions, working memory, and postural balance in both healthy and cognitively impaired older adults [16,149]. Dual tasking efficacy is particularly enhanced in individuals with declines in vision, proprioception, and vestibular function [150].

The study had several limitations. First, no risk of bias or article quality assessment was performed due to methodological heterogeneity. Second, the effect sizes were not reported. The SOP was based only on parameter frequency, without considering the study sample sizes. Narrative synthesis was used, as a meta-analysis was not feasible due to heterogeneity in the secondary tasks and outcomes measures, limiting the strength of the conclusions. Outcomes were considered in general, without categorization, to maintain focus on the training protocols. Training volume variability made statistical analysis unfeasible.

Future studies adopting common protocols could help in quantifying the real impact of DT training, and evaluate outcomes more rigorously. This review did not address the use of technology to ensure SOP applicability across settings. Even if studies exist on the use of exergames as cognitive training [151], this exercise methodology was also excluded due to its distinct mechanisms. However, technology-based DT interventions may be effective on global cognitive function, inhibitory control, and processing speed [123]. Future research should include original field studies and randomized controlled trials directly comparing different types of cognitive tasks (e.g., spatial memory vs. calculation) combined with the same motor tasks. It would be also interesting to study DT training as a tool for the performance of fine motor skills such as manual dexterity. Indeed, different populations suffer from hand tremors, and one recent study investigated the correlation between hand exercise and hand tremors [152], so its understanding in a DT contest could also provide more information. It would also be valuable to create population-specific SOPs, particularly for those with cognitive disorders, considering the DT training role in the prevention and deceleration of cognitive impairments [153]. Standardized trials should better assess the progression, modulation, and task difficulty, tailored to participant characteristics. Only then will true progress in SOP development be achieved. Our SOP aims to highlight the need for greater clarity in this emerging field.

## 5. Conclusions

The topic of DT training for older adults presents a diverse and heterogenous literature. Comparisons between protocols were limited by the variability of the articles, compromising the overall quality of the research. The SOP proposed for healthy older adults consists of two sessions per week, each lasting 60 min, with adapted and progressively intensified exercises. Postural balance, resistance, aerobic, and mobility training are included. The secondary tasks proposed are primarily mental tracking or working memory tasks. The SOP is intended as a starting point, with the hope that future studies will build upon and refine this direction.

## Figures and Tables

**Figure 1 brainsci-15-00785-f001:**
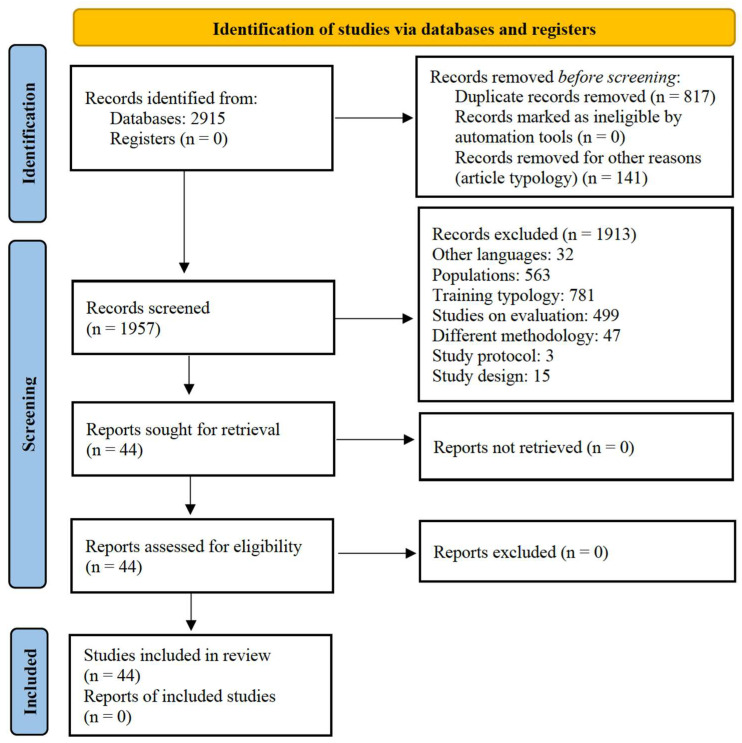
PRISMA 2020 flow diagram for the scoping reviews, which only included searches of the databases and registers.

**Table 1 brainsci-15-00785-t001:** Articles overview.

1st Author and Year	N and Sample Age	Duration, Frequency, Time	Intensity/Type/Protocol	Adverse Events/Adherence and Dropout
Akin 2021 [44]	50, 68 yr	8-weeks, 40-min, G	10 min WU, 20 min C, 10 min CD	No dropout
Amini 2022 [45]	42, 70 yr	4-weeks, 2/week, 45-min	I: progressive. ACSM, protocol [46]	No dropout
Anandh 2018 [43]	96, 65–75 yr	3 ses of training	10 min WU, walking: 5 min	NI
Ansai 2017 [47]	80, 68 yr	12-weeks, 3/week, 50-min, G	10 min WU, 30 min C, 10 min S. ACSM [48], AGSPEO [49]	Training adherence: >70%
Machado Araújo 2025 [34]	11, 83 yr	12-weeks, 2/week, small G	I: progressive, %1 RM, protocol [50,51]	Dropout rates: <25%
Arcila Castaño 2022 [51]	30, 70 yr	16-weeks, 2/week, G	I: progressive, %1 RM	Dropout rates: <20%
Bagheri 2021 [52]	70, 65 yr	16 ses, 3/week, 60-min	I: progressive-adapted	Dropout rates: <12%
Bischoff 2021 [53]	24, 83 yr	16-weeks, 2/week, 45–60-min	5–10 min WU-mobilization, 40 min C, 5–10 min CD. I: progressive. ACSM [54], protocol [55,56,57,58,59,60,61,62,63,64]	Dropout rates: <4%; Training adherence: >75%
Blasco-Lafarga 2020 [65]	25, 70 yr	Protocol [66], G	5–10 min CD (amusing, social tasks). I: adapted. Protocol [66]	Dropout rates: <26%
Chen 2023 [67]	50, 68 yr	12-weeks, 2/week, 90-min, G	10–15 min WU, 60 min C, 15–20 min S. I: progressive. ACSM [68]	NI
Chuang 2024 [35]	158, 74 yr	12 ses, 1/week, 120-min	NI	Dropout rates: <12%
Conradsson 2019 [69]	95, 76 yr	12-weeks, 3/week, 45-min, G	5 min WU (walk), 30 min C, 5 min CD (walk, S). I: progressive-adapted. Protocol [70]	NI
de Maio Nascimento 2023 [71]	50, 66 yr	12-weeks, 2/week, 60-min	10 min WU, 40 min C, 10 min relax. I: progressive. Protocol [72]	Training adherence: >75%
Falbo 2016 [73]	36, 72 yr	12-weeks, 2/week, 60-min, G	10 min WU, 30 min C, 20 min S, relax. I: adapted. Protocol [74]	Dropout rates: <36%
Granacher 2021 [75]	51, 66 yr	8-weeks, daily, 2 per day	I: progressive. Protocol [76]	Training adherence: >92%
Gregory 2016 [77]	44, 73 yr	26-weeks, 2–3/week, 60–75-min, G–I	I: target heart rate (70–85% max HR). Protocol [78]	Training adherence: >78%
Halvarsson 2015 [79]	69, 76 yr	12-weeks, 3/week, 45-min, G	I: 3 levels (basic, moderate, advanced), progressive	Training adherence: >89%
Halvarsson 2016 [80]	96, 76 yr	12-weeks, 3/week, 45-min, G	I: progressive. Protocol [70]	Dropout rates: <12%
Ho 2023 [81]	136, 75 yr	12-weeks, 2/week, 60-min	Protocol [82]	NI
Horata 2021 [83]	32, 65 yr	6-weeks, 2/week, 60-min, I	10 min WU, 10 min CD (breathing, S). I: progressive-adapted. Protocol [84]	Dropout rates: <16%
Jabeen 2024 [36]	30, 62 yr	6-weeks, 3/week, 40-min	NI	No dropout
Jardim 2021 [85]	72, 67 yr	24 ses, 2/week, 75-min, G	10 min WU, 60 min C, 5 min S. I: moderate (60–70% of max HR). Protocol [54]	Dropout rates: <22%
Kim 2019 [86]	20, 79 yr	8-weeks, 2/week, 30-min, I	I: adapted	NI
Kitazawa 2015 [87]	60, 76 yr	8-week, 1/week, 60-min	I: increased	No dropout
Konak 2016 [88]	42, 68 yr	4 weeks, 3/week, 45-min, I	NI	Dropout rates: <11%
Mundada 2022 [89]	40, 64 yr	6-weeks, 3/week, 45-min	NI	No dropout
Nieto-Guisado 2024 [90]	20, 73 yr	30-min	5 min WU, 20 min C, 5 min CD. Mobility, dance	NI
Nishiguchi 2015 [41]	48	12-weeks, 1/week, 90-min, G	15 min S, moderate-intensity, 15 min strength, 60 min DT. I: increased. ACSM [91]	Training adherence: >91%
Norouzi 2019 [92]	60, 68 yr	60–80-min	Resistance training	No dropout
Pantoja-Cardoso 2023 [93]	62, 67 yr	16-weeks, 3/week	WU (joint mobility, basic movements), C, CD (breathing, S). I: adapted based on RPE	Dropout rates: <12%
Párraga-Montilla 2021 [94]	43, 81 yr	8-weeks, 5/week, 60-min	10 min WU, 40 min C, 10 min CD. I: increased	No dropout
Rajalaxmi 2022 [37]	50, 65–85 yr	12-weeks, 5/week	NI	NI
Rezola-Pardo 2019 [38]	85, 85 yr	12-weeks, 2/week, 60-min	I: moderate, adapted. Protocol [95,96]	Dropout rates: <17%
Rezola-Pardo 2022 [39]	85, 85 yr	12-weeks, 2/week, I	5 min WU, 5 min breath-relax. I: individualized. Protocol [38]	Dropout rates: <17%
Rosado 2021 [97]	51, 75 yr	24-weeks, 3/week, 75-min	10 min WU, 50 min C, 5 min CD (S or breath), 5 min begin-finish ritual. I: moderate intensity (~13 point, RPE)	Dropout rates: <14%; Training adherence: >82%
Santos 2023 [98]	36, 83 yr	12-weeks, 2/week, 50-min	6 WU. Protocol [99]	Dropout rates: <25%
Sedaghati 2022 [100]	28, 70 yr	8-weeks, 3/week, 60-min	Protocol [95,96,101,102,103]	No dropout
Sok 2021 [104]	65, 74 yr	10-weeks, 2/week, 50-min, G	5 min WU, 60 min C, 5 min CD, 10 min communication time. I: adapted-increased. Protocol [105]	NI
Tait 2025 [106]	96, 77 yr	24-weeks, 2/week, 45–60-min, G	WU, 2–3 C, CD. I: moderate-hard on RPE. Progressive	Training adherence: >58%
Trombini-Souza 2023 [42]	60	24-weeks, 2/week, 60-min G	10 min WU (walk, S, joint mobilization), 40 C; 10 min relax (breathing, S. Protocol) [107]	Dropout rates: <50%
Wollesen 2017 [62]	95, 71 yr	12-weeks, 1/week, 60-min, G	NI	Dropout rates: <11%
Wollesen 2017 [56]	90, 72 yr	NI	I: increased. Protocol [63]	Dropout rates: <4%
Yokoyama 2015 [108]	27, 74 yr	12-weeks, 3/week, 60-min	15 min of mental gymnastics (finger motions), 35 C, 10 min of flexibility exercise. I: increased	Dropout rates: <7%; Training adherence: >90%
Yuzlu 2022 [40]	58, 85 yr	8-weeks, 2/week, I	10 min WU, 40 min C, 10 min CD	Dropout rates: <14%

American College of Sports Medicine: ACSM; American Geriatrics Society Panel on Exercise and Osteoarthritis: AGSPEO; cool-down: CD; conditioning: C; dual task: DT; group: G; heart rate: HR; individual: I; no info: NI; percentage of 1 repetition maximal: %1 RM; rate of perceived effort: RPE; sessions: ses; stretching: S; years: yr; warm-up: WU.

**Table 2 brainsci-15-00785-t002:** Number of studies for the parameters of protocol duration, session number, and duration.

Protocol Duration (Weeks)	No. of Studies	Session a Week (n)	No. of Studies	Duration (Minutes)	No. of Studies	Intensity	No. of Studies
12	16	2	18	60	12	Progressive	15
8	7	3	12	45	6	Adapted	4
24	3	1	4	50	3	PA	4
16	3	5	2	30	2		
6	3	7	1	40	2		
4	2	2–3	1	75	2		
10	1			90	2		
26	1			120	1		

Note: number: No; progressive-adapted: PA.

**Table 3 brainsci-15-00785-t003:** Number of studies for aspects evaluated in the included studies.

Aspects Evaluated	No.	Aspects Evaluated	No.	Aspects Evaluated	No.	Aspects Evaluated	No.
Cognitive function	20	Quality of life	2	Psycho-affective status	1	oxidative stress	1
Postural balance	15	Functional ability	1	Depression	1	Inflammation	1
Walking or gait analysis	11	Brain activation efficiency	1	Life satisfaction	1	Vascular health	1
Physical fitness	10	Executive functions	1	Frailty	1	Brain-derived neurotrophic factor	1
Falls and risk of falls	6	Dual-task interference	1	Activity of Daily Living	1	Circulating neurological	1
Muscle strength	3	Dual-tasking	1	Psychosocial wellbeing	1	Inflammatory markers	1
Cognition	3	Health status	1	Renal function	1	Plasma amyloid β peptide	1
Mobility	2	Cognitive health	1	Lipid profile	1	Adherence to programs	1

Note: number: No.

**Table 4 brainsci-15-00785-t004:** Training characteristics, variables considered, and effects obtained.

1st Author and Year	Primary Task	Secondary Task	Primary Variables Evaluated	Primary Effect
Akin 2021 [44]	MT, D	Mental tracking, MT	PB, fear of fall, walk, strength	++
Amini 2022 [45]	MT, D	Working memory, reaction, discrimination-decision making, mental tracking, verbal fluency [46]	Cognitive health components	++
Anandh 2018 [43]	MT, walk	Verbal fluency, reaction, other (talking)	PB, gait	++
Ansai 2017 [47]	MT, RT, PB	Mental tracking, verbal fluency, other	CF, PF	ND
Machado Araújo 2025 [34]	MT, RT	Verbal fluency [50,51]	PF, CF, lipid profile, renal function, oxidative stress, inflammation	++
Arcila Castaño 2022 [51]	MT, RT	Verbal fluency	Body composition, PF, CF, plasma brain-derived neurotrophic factor	++
Bagheri 2021 [52]	MT, S-D PB	Verbal fluency, mental tracking	dual-task interference	++
Bischoff 2021 [53]	D PB, AT, RT	Verbal fluency, working memory, discrimination-decision making, mental tracking; MT	PF, psychosocial well-being	++
Blasco-Lafarga 2020 [65]	DLA	Protocol (Spanish) [66]	PF, CF	++
Chen 2023 [67]	AT	Working memory, discrimination-decision making, reaction, verbal fluency, mental tracking [109]; MT	CF, functional fitness	++
Chuang 2024 [35]	Stretching, RT, AT, PB	Sequential and simultaneous cognitive and physical training. Working memory, mental tracking, verbal fluency, other	Instrumental activities of daily living	++
Conradsson 2019 [69]	S-D PB	Mental tracking, working memory; MT	Gait, PB	++
De Maio Nascimento 2023 [71]	Gait, PB [72]	Mental tracking, verbal fluency, working memory, other	PB, gait, lower limb strength, CF	++
Falbo 2016 [73]	DLA, mobility	Discrimination-decision making, working memory [110]	Executive CF, gait	++
Granacher 2021 [75]	PB	Other (tooth brushing)	PB, strength	ND
Gregory 2016 [77]	Stepping [111]	Verbal fluency, mental tracking	Gait, vascular health	++
Halvarsson 2015 [79]	S-D PB	Cognitive, MT [70]	Fall-related self-efficacy, fear of falling, gait, PB, PF	++
Halvarsson 2016 [80]	S-D PB	Cognitive, MT [70]	Fall	++
Ho 2023 [81]	Protocol [82]	No details	functional ability	++
Horata 2021 [83]	Gait, PB	Working memory [84]	Gait, cognition	++
Jabeen 2024 [36]	MT	Verbal fluency, working memory, mental tracking	PB, quality of life	++
Jardim 2021 [85]	AT, RT, PB, coordination	MT; verbal fluency, working memory, mental tracking, discrimination-decision-making, other [112]	CF, PF	++
Kim 2019 [86]	MT (walk)	MT [113,114]	CF, gait	++
Kitazawa 2015 [87]	Stepping	Other (rhythmic tasks)	CF, gait, adherence to programs	++
Konak 2016 [88]	PB	Mental tracking, verbal fluency	PB, activity-specific PB confidence	++
Mundada 2022 [89]	S-D MT	Verbal fluency, mental tracking	CF	++
Nieto-Guisado 2024 [90]	PB	Other	PB, knee proprioception	ND
Nishiguchi 2015 [41]	Stepping	Verbal fluency, decision making, reaction [115,116]	CF, brain activation efficiency	++
Norouzi 2019 [92]	RT	MT; working memory, mental tracking [117,118]	CF, PB	++
Pantoja-Cardoso 2023 [93]	S-D PB, coordination, mobility, MT	Working memory, mental tracking, reaction	CF	+
Párraga-Montilla 2021 [94]	MT	Mental tracking, working memory, verbal fluency	CF, PF	++
Rajalaxmi 2022 [37]	S-D PB	Reaction, working memory; MT [119,120]	PB, CF	++
Rezola-Pardo 2019 [38]	RT, PB	Working memory, discrimination-decision making [103]	DT, PF, CF, psycho-affective status, quality of life, frailty	ND
Rezola-Pardo 2022 [39]	RT, S-D PB	Protocol [103]	Number of falls, number of fallers, parameters associated with fall risk	-
Rosado 2021 [97]	Agility, body awareness, mobility, RT	Working memory, discrimination-decision making	CF, mobility, DT	++
Santos 2023 [98]	Protocol [99]	Verbal fluency, working memory, mental tracking	Risk of falling	++
Sedaghati 2022 [100]	RT, PB [95,96,101,102,103]	Working memory, verbal fluency, mental tracking	PB	++
Sok 2021 [104]	Stepping [105]	Other (songs and rhythms)	CF, health status, depression, life satisfaction	++
Tait 2025 [106]	PB/mobility, RT	Mental tracking, working memory; MT	Cognition, circulating neurological, inflammatory markers	+
Trombini-Souza 2023 [42]	Walk, PB	Mental tracking, working memory, verbal fluency [107]	Mobility, CF, PB	++
Wollesen 2017 [62]	DLA, walk [22]	Reaction; MT	Walk	++
Wollesen 2017 [56]	DLA, walk [63]	Working memory	Gait	++
Yokoyama 2015 [108]	RT, AT	Mental tracking, other (Shiritori)	Executive functions, plasma amyloid β peptide (Aβ) 42/40 ratio	++
Yuzlu 2022 [40]	Motor skills, PB	Working memory, mental tracking, reaction [6,121]	PB, fear of falling, gait	ND

Aerobic training: AT; cognitive function: CF; daily life activities: D; dynamic: D; important effect: ++; moderate effect: +; motor task: MT; negative effect: -; no differences: ND; physical function: PF; postural balance: PB; resistance training: RT; static: S.

**Table 5 brainsci-15-00785-t005:** Number of studies for each parameter of the primary and secondary tasks.

Motor Tasks	No. of Studies	Cognitive Tasks	No. of Studies
Postural balance	21	Mental tracking	23
Resistance training exercises	13	Working memory tasks	22
Gait and walking exercises	6	Verbal fluency tasks	20
Aerobic training	6	Motor task	12
Stepping	4	Reaction time tasks	8
Mobility exercise	4	Discrimination-decision making	7
Daily life activities	4		
Coordination exercises	2		
stretching	1		
Agility/body awareness	1		
motor skills	1		

**Table 6 brainsci-15-00785-t006:** Standard operating procedure for a dual-task training in healthy older adults.

Frequency:	2 sessions a week for 12-weeks	
Intensity:	Progressive-adapted (rate of perceived effort)	
Time:	60 min	
Type: Dual task		
Primary task	Postural balance	Static and dynamic exercises
	Resistance training	Machine-based and power training
	Aerobic training	High-intensity interval or continuous moderate-intensity
	Mobility	Static training
	Daily life activities	
Secondary task	Mental tracking	Counting backward, math calculations, solving anagrams
	Working memory (long- and short-term)	Remembering words, items, shapes, colors, or numbers, complementing proverbs or traditional songs
	Verbal fluency	Reading and talking, reverse spelling, and singing songs
	Motor	Holding glasses, carrying and/or manipulating objects, tapping, throwing or holding objects, balancing a cup on the palm

## Data Availability

Not applicable.

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
