# Peer review of "A Standard Operating Procedure for Dual-Task Training to Improve Physical and Cognitive Function in Older Adults: A Scoping Review"

_brainsci, 2025, doi:10.3390/brainsci15080785_

Round 1

Reviewer 1 Report

Comments and Suggestions for Authors

This systematic review addresses a topic of significant clinical and practical importance. With the global population aging, developing effective interventions to delay the decline of physical and cognitive function in older adults is crucial. Dual Task Training (DTT) is a promising field of research, and the manuscript's aim to establish a Standard Operating Procedure (SOP) for such training is both ambitious and challenging. The manuscript is structurally complete and provides a degree of summarization of existing research. However, there is substantial room for improvement in its argumentative rigor, methodological transparency, presentation of results, and depth of discussion. In its current form, the manuscript reads more like a narrative literature review than a systematic review capable of proposing an SOP.

Major Comments

  1. There is a significant disconnect between the manuscript's stated objective and its actual content. The title and abstract explicitly promise to establish a "Standard Operating Procedure (SOP)." However, upon reading the full text, the paper primarily summarizes the diversity in existing DTT protocol designs and even concludes by suggesting that an SOP is needed in the future. This contradicts the paper's original premise. The authors must make a choice: either revise the paper's objective and title to clearly position it as "a systematic review describing the heterogeneity of DTT protocols," or substantively synthesize the evidence in the results and discussion sections to propose a concrete, actionable SOP framework or detailed recommendations.
  2. The methodological rigor and transparency are insufficient. For a systematic review, the methods section is paramount. Although the authors mention the PRISMA guidelines, they have not fully adhered to its reporting standards. For example, the provided PRISMA flow diagram is oversimplified and fails to state the specific reasons for excluding studies at each screening stage. Regarding the risk of bias assessment, the authors used the PEDro scale but did not report the detailed results for each study in the results section, nor did they analyze the impact of this risk of bias on the overall strength of the evidence in the discussion.
  3. The presentation of results lacks logical structure and analytical depth. The current results section reads like a list of information, merely cataloging the training parameters from different studies. This format has poor readability and makes it difficult for readers to identify patterns or draw meaningful conclusions. I strongly recommend that the authors reorganize the results using structured tables and figures and that they classify and synthesize the findings rather than simply providing a descriptive list.

Specific Comments by Section

Abstract

  1. The background information in the abstract is too elementary. Sentences like, “The dual task (DT) concept is based on the performance of two concurrent tasks,” should be replaced to directly address the core problem of the current lack of standardized protocols for DTT research.
  2. The description of the research objective is unacademic. The phrasing, “The DT training wants to provide…,” should be revised into more rigorous and formal scientific language to state the review’s purpose.
  3. The results section of the abstract needs to be more specific. Instead of merely stating that “a standard operating procedure has been proposed,” it should briefly summarize the core components of this SOP to reflect the paper's actual contribution.
  4. The conclusion of the abstract contains a logical contradiction. Stating that “a SOP is needed” is self-defeating for a systematic review that aims to provide one. The conclusion should summarize the key evidence this paper has provided or the framework it has proposed for establishing an SOP.

Introduction

  1. The opening of the introduction is overly broad. It is recommended not to start from the general concept of “aging” but to focus directly on the dual challenge of concurrent physical and cognitive decline in older adults, thereby introducing the decline in dual-task performance as a key manifestation of this challenge.
  2. The theoretical background on dual-tasking could be deepened. It is advisable to integrate neuroscientific evidence, such as the role of the prefrontal cortex in dual-task processing, to better support the theoretical basis of DTT.
  3. When distinguishing between “dual task as a test” and “dual task as a training,” the authors could use examples from key studies to illustrate how DT assessment reveals risks, thereby more forcefully emphasizing the necessity of DTT.
  4. The end of the introduction needs to articulate the "research gap" more clearly and specifically. It is not enough to say “no guidelines exist”; the authors should detail the problems this absence causes, such as difficulties in comparing study results or a lack of guidance for clinical application.
  5. The research objective stated at the end of the introduction must align with the paper's actual output. It would be best to provide a clear, operational definition of what a “Standard Operating Procedure (SOP)” entails in the context of this paper.

Methods

  1. The authors need to provide a complete example of their search strategy in an appendix, detailing all keywords, Boolean operators, and filters used in at least one major database (e.g., PubMed) to ensure the study's replicability.
  2. The definitions for inclusion and exclusion criteria need to be more precise. For example, the specific age range for “Older adults” (e.g., ≥60 or ≥65 years) must be specified, and an operational definition for “Dual Task Training” must be provided.
  3. More information is needed in the data extraction phase. I suggest the authors consider extracting and reporting on training-related “adverse events,” which is critical for assessing the safety of DTT.
  4. Similarly, in the data extraction section, I recommend adding the extraction and analysis of data on study “adherence” and “dropout rates,” as this directly relates to the feasibility of the training protocols in real-world settings.
  5. When reporting the risk of bias assessment, a chart (such as the traffic light plot or bar chart recommended by Cochrane) should be used to clearly display the specific scores of each study on each item of the PEDro scale, rather than just providing a total score.
  6. The authors state in the methods that a “Narrative synthesis” will be conducted. Therefore, they need to explicitly describe the methodological framework they are following for this synthesis to demonstrate that their analysis is systematic rather than descriptive.

Results

  1. The PRISMA flow diagram needs to include more detail. In the “Records excluded” step, the primary reasons for exclusion and the corresponding number of studies must be listed.
  2. A column could be added to Table 1 in the results section to briefly describe the Primary Outcomes measured in each study. This would allow readers to quickly grasp the focus of each study.
  3. The core content of the results—the description of the interventions—should not be presented in long paragraphs of text. It is strongly recommended that this information be fully tabulated. The authors could design several large tables to detail the motor tasks, cognitive tasks, and training dosage specifics.
  4. After listing the details of all studies, summary paragraphs must be added. For example, the authors should synthesize information such as, “Across the XX included studies, the most common motor task was walking (XX%), and the most common cognitive task was arithmetic calculation (XX%).”
  5. The manuscript currently completely lacks a systematic summary of the Outcome Measures used across the studies. A section must be added to categorize and summarize the specific tools used to assess physical and cognitive function.

Discussion

  1. The first paragraph of the discussion should not simply repeat the results but should be a concise synthesis of the findings, pointing out the core characteristics of current DTT research, such as the significant uncertainty and heterogeneity in its protocol design.
  2. The discussion needs to compare this review's findings with previous systematic reviews in the field, explicitly stating whether the findings support, supplement, or challenge prior conclusions.
  3. This is the section most in need of strengthening. The authors must systematically explore the key issues for constructing an SOP, including task selection, principles of intensity and progression, and training dosage.
  4. Based on the foregoing in-depth discussion, the authors should attempt to propose a preliminary, modular SOP framework in the discussion section, demonstrating how the review's findings can be translated into practical guidance.
  5. To increase the theoretical depth of the discussion, the authors could delve deeper into the potential neural mechanisms of DTT efficacy, such as neural plasticity and brain network reorganization, and link these theories to the effective training elements observed in the results.
  6. The limitations analysis needs to be more in-depth. In addition to mentioning publication bias, the authors should frankly discuss how the significant heterogeneity among the included studies prevented a quantitative synthesis, thereby limiting the strength and generalizability of the conclusions.
  7. The directions for future research need to be very specific. For example, recommend that “future RCTs should directly compare the effects of different types of cognitive tasks (e.g., spatial memory vs. calculation) when combined with the same motor task,” rather than speaking in generalities.

Other Details

  1. The entire manuscript has several grammatical errors and typos. It must be thoroughly proofread by a native English speaker or a professional editing service before resubmission.
  2. The authors should carefully check the reference list to ensure that every citation is formatted strictly according to the journal's guidelines.

Author Response

This systematic review addresses a topic of significant clinical and practical importance. With the global population aging, developing effective interventions to delay the decline of physical and cognitive function in older adults is crucial. Dual Task Training (DTT) is a promising field of research, and the manuscript's aim to establish a Standard Operating Procedure (SOP) for such training is both ambitious and challenging. The manuscript is structurally complete and provides a degree of summarization of existing research. However, there is substantial room for improvement in its argumentative rigor, methodological transparency, presentation of results, and depth of discussion. In its current form, the manuscript reads more like a narrative literature review than a systematic review capable of proposing an SOP.

Reply: thank you very much for the comment, the time and effort spent in our manuscript. It is really appreciated. All the changes are highlighted in yellow within the manuscript and a point-to-point reply is provided below.

Major Comments

  1. There is a significant disconnect between the manuscript's stated objective and its actual content. The title and abstract explicitly promise to establish a "Standard Operating Procedure (SOP)." However, upon reading the full text, the paper primarily summarizes the diversity in existing DTT protocol designs and even concludes by suggesting that an SOP is needed in the future. This contradicts the paper's original premise. The authors must make a choice: either revise the paper's objective and title to clearly position it as "a systematic review describing the heterogeneity of DTT protocols," or substantively synthesize the evidence in the results and discussion sections to propose a concrete, actionable SOP framework or detailed recommendations.

Reply: thank you for this valuable comment. We have revised parts of the paper and made a significant effort to draw the reader’s attention to the synthesis of the evidence presented in the Results section, which supports the rationale behind our proposed SOP. Furthermore, the Results section has been enhanced with tables to better highlight key findings. The Discussion section has been also partially rewritten and restructured to more clearly explain the logic and relevance of our approach.

We hope that the work undertaken is appreciated and satisfactorily addresses this comment. However, if the Reviewer believes that further actions are needed, we will be more than happy to make additional improvements to the manuscript.

  1. The methodological rigor and transparency are insufficient. For a systematic review, the methods section is paramount. Although the authors mention the PRISMA guidelines, they have not fully adhered to its reporting standards. For example, the provided PRISMA flow diagram is oversimplified and fails to state the specific reasons for excluding studies at each screening stage. Regarding the risk of bias assessment, the authors used the PEDro scale but did not report the detailed results for each study in the results section, nor did they analyze the impact of this risk of bias on the overall strength of the evidence in the discussion.

Reply: Thank you very much for this comment. It is extremely useful and appreciated. After this first revision round, and because another reviewer highlighted that the methodology is not sufficient for our manuscript to consider it a systematic review, we decided to adopt PRISMA guidelines for scoping reviews. This decision was made because the manuscript’s methodology cannot be modified to include risk of bias article quality assessments, and it more appropriately aligns with the aims of a scoping review.

The flow diagram used was the one provided by PRISMA, and we followed its structure step by step. To our knowledge, the structure of the PRISMA flow diagram cannot be altered, but in response to the Reviewer’s request, we have modified it as suggested.

Regarding the risk of bias and quality assessment, the included articles are too heterogeneous, making their evaluation difficult. The variety of articles types would require different assessment tools, complicating direct comparisons of the results.

  1. The presentation of results lacks logical structure and analytical depth. The current results section reads like a list of information, merely cataloging the training parameters from different studies. This format has poor readability and makes it difficult for readers to identify patterns or draw meaningful conclusions. I strongly recommend that the authors reorganize the results using structured tables and figures and that they classify and synthesize the findings rather than simply providing a descriptive list.

Reply: The results have been partially rewritten, and new Tables have been added. The existing tables were also partially modified. In doing so, we aimed to better explain the rationale behind our SOP and to make the text easier to read and to follow. Thanks to the valuable feedback from the Reviewers, our goal was to guide the reader toward our conclusions in a clear and understandable way. If this goal has not been fully achieved, we kindly ask the Reviewer to provide additional feedback or specific comments so we can further address this point. However, we hope that our efforts are appreciated, given the substantial work done in this regard.

Specific Comments by Section

Abstract

  1. The background information in the abstract is too elementary. Sentences like, “The dual task (DT) concept is based on the performance of two concurrent tasks,” should be replaced to directly address the core problem of the current lack of standardized protocols for DTT research.

Reply: Thank you for this suggestion. We have removed the sentence requested bringing the reader directly to the DT training and its limitations.

  1. The description of the research objective is unacademic. The phrasing, “The DT training wants to provide…,” should be revised into more rigorous and formal scientific language to state the review’s purpose.

Reply: This section of the abstract has been edited, retaining the essential elements of the Methods section. We hope that it now appears more rigorous and written in a formal scientific language.

  1. The results section of the abstract needs to be more specific. Instead of merely stating that “a standard operating procedure has been proposed,” it should briefly summarize the core components of this SOP to reflect the paper's actual contribution.

Reply: Thank you for the comment. In accordance with the reviewer’s indications, we revised this section of the abstract, aiming to summarize the core components of our SOP.

  1. The conclusion of the abstract contains a logical contradiction. Stating that “a SOP is needed” is self-defeating for a systematic review that aims to provide one. The conclusion should summarize the key evidence this paper has provided or the framework it has proposed for establishing an SOP.

Reply: Thank you for the comment. As suggested , we have edited this section of the abstract and removed the contradictory statement identified by this Reviewer.

Introduction

  1. The opening of the introduction is overly broad. It is recommended not to start from the general concept of “aging” but to focus directly on the dual challenge of concurrent physical and cognitive decline in older adults, thereby introducing the decline in dual-task performance as a key manifestation of this challenge.

Reply: We restructured the first part of the introduction, removing some sentence about the aging process. We hope that we have addressed this issue, but if not, we kindly ask the Reviewer to indicate the relevant part with line and page numbers. Thank you.

  1. The theoretical background on dual-tasking could be deepened. It is advisable to integrate neuroscientific evidence, such as the role of the prefrontal cortex in dual-task processing, to better support the theoretical basis of DTT.

Reply: Thank you for this suggestion. As requested, we added some sentences about the relationship between dual-task processing and the prefrontal cortex. We hope the Reviewer appreciates the added section and the references cited: “This training methodology takes different forms and can be adapted to participant’s level and context [21]. The advantage of DT training lies in its benefits for both motor and cognitive functions. Indeed, this methodology significantly engages with the left prefrontal cortex and the parietal area [22]. In young adults, DT situations have been shown to increase activity in the prefrontal cortex [23]. This brain region is considered an executive hub, and it is strongly associated with executive function [24]. Consequently, DT training improves executive function more effectively than single cognitive training [25,26]”

  1. When distinguishing between “dual task as a test” and “dual task as a training,” the authors could use examples from key studies to illustrate how DT assessment reveals risks, thereby more forcefully emphasizing the necessity of DTT.

Reply: Thank you for this useful comment. We added the following sentences in the manuscript: “DT testing aims to reveal individual risks through the concurrent combination of a primary task (such as a postural balance test or a walking test) and a secondary task (such as a manual or a cognitive task). Examples include studies on postural balance [10] and gait ability [11,12], which are also used to predict possible future falls in healthy [13,14], and pathological populations [15]. This testing methodology is also used to assess cognitive performance [16], influence manual dexterity [17], and evaluate mobility [18,19]. Another important example of DT testing is its use as a predictive tool for cognitive impairment [20].

Considering the various domains in which the DT concept is applied, it also shows strong potential as a training tool. Consequently, greater attention should be given to DT training.”.

  1. The end of the introduction needs to articulate the "research gap" more clearly and specifically. It is not enough to say “no guidelines exist”; the authors should detail the problems this absence causes, such as difficulties in comparing study results or a lack of guidance for clinical application.

Reply: Thank you for this comment, we highlighted this part in the text and it is reported here: “Unfortunately, DT training includes diverse and inconsistent protocols, limiting comparisons between studies [35]. A standardized DT training protocol does not exist [30], and there is considerable heterogeneity in this field [33]. The limitation research gap that still exists in this training methodology lies in  the variety of approaches and, at times, the impracticality of applying them in everyday contexts [21]. A review highlighted that gaps still exist regarding the optimal dose and best approach to adopt [32], and there is also a lack of reporting on key implementation details [21]. Furthermore, it would be valuable to study the most effective training protocols tailored for specific diagnostic groups [35]”

  1. The research objective stated at the end of the introduction must align with the paper's actual output. It would be best to provide a clear, operational definition of what a “Standard Operating Procedure (SOP)” entails in the context of this paper.

Reply: Thank you for this comment, we integrated a part in the text, now it is presented as: “Standard operating procedures (SOP) are present in the literature and are defined as detailed documents outlining each step of a procedure [37,38]. They are also used in the sports science field, providing guidelines and ensuring that procedures are replicable and comparable [39].”.

Methods

  1. The authors need to provide a complete example of their search strategy in an appendix, detailing all keywords, Boolean operators, and filters used in at least one major database (e.g., PubMed) to ensure the study's replicability.

Reply: Thank you for this point. Because we avoided the use of filters, we simply copied and pasted the string in the three databases, we prefer not add additional material in the appendix. We added a sentence in the text to make our working procedure clearer: “The string was adopted in the three databases in the current form. No filters were used.”

  1. The definitions for inclusion and exclusion criteria need to be more precise. For example, the specific age range for “Older adults” (e.g., ≥60 or ≥65 years) must be specified, and an operational definition for “Dual Task Training” must be provided.

Reply: Thank you for this question. In the text, after “older adults”, in the eligibility criteria section, it was specified as 65 years or above for the range for “older adults”.

We also added a sentence to better provide an operational definition for “dual task training”: “The training had to include a primary task performed with another task, called secondary task. The nature of the primary or the secondary task was not an eligibility criteria.” We hope the Reviewer appreciate this solution.

  1. More information is needed in the data extraction phase. I suggest the authors consider extracting and reporting on training-related “adverse events,” which is critical for assessing the safety of DTT.

Reply: Thank you for this suggestion, we added it in the text as: “the adverse events, adherence, and dropout rates”

  1. Similarly, in the data extraction section, I recommend adding the extraction and analysis of data on study “adherence” and “dropout rates,” as this directly relates to the feasibility of the training protocols in real-world settings.

Reply: Thank you for this suggestion, we added it in the text as: “the adverse events, adherence, and dropout rates”

  1. When reporting the risk of bias assessment, a chart (such as the traffic light plot or bar chart recommended by Cochrane) should be used to clearly display the specific scores of each study on each item of the PEDro scale, rather than just providing a total score.

Reply: Thank you for this comment. Because, according to two Reviewers, our methodology is not sufficient to classify the manuscript as a systematic review, as a scoping review, the risk-of-bias assessment is not required.

  1. The authors state in the methods that a “Narrative synthesis” will be conducted. Therefore, they need to explicitly describe the methodological framework they are following for this synthesis to demonstrate that their analysis is systematic rather than descriptive.

Reply: Thank you for this comment. Because the systematic approach of our work was highlighted as a major limitation of the manuscript, we chose to classify it as a scoping review, allowing us to describe the topic without a systematic analysis. Furthermore, the narrative synthesis was performed following two principles. The first was the presentation of the SOP based on the frequency of the parameters evaluated. The second, was the discussion of the SOP with the current literature. We hope that now, our decision to proceed in this way is appreciated.

Results

  1. The PRISMA flow diagram needs to include more detail. In the “Records excluded” step, the primary reasons for exclusion and the corresponding number of studies must be listed.

Reply: we modified the PRISMA flow diagram as requested, trying to retain the original format at the same time.

  1. A column could be added to Table 1 in the results section to briefly describe the Primary Outcomes measured in each study. This would allow readers to quickly grasp the focus of each study.

Reply: Thank you for this comment. The column of Table 1 “objective of the study” has been changed to “primary variables evaluated”, while the column “effect” of Table 2 has been changed to “primary outcomes”. We included both columns in the same table (Table 2). We hope that  with this wording, readers will quickly grasp the focus of each study.

  1. The core content of the results—the description of the interventions—should not be presented in long paragraphs of text. It is strongly recommended that this information be fully tabulated. The authors could design several large tables to detail the motor tasks, cognitive tasks, and training dosage specifics.

Reply: Three tables have been added, Table 2, Table 4 and 5 to try to summarize the results in a clearer and more visually appearing way. The second part of paragraph 3.3 was only partially changed because it contains information that is not easily included withing a table. We think this content offers valuable insights that readers could use as alternative strategies for intervention. We hope the Reviewer appreciates our effort. Any suggestions are well appreciated.

  1. After listing the details of all studies, summary paragraphs must be added. For example, the authors should synthesize information such as, “Across the XX included studies, the most common motor task was walking (XX%), and the most common cognitive task was arithmetic calculation (XX%).”

Reply: As requested, the text has been simplified, and some parts have been removed. We hope that this section is now clearer and easier to read. The percentage, unfortunately, cannot be used because some secondary or primary tasks are proposed in more than one study.

  1. The manuscript currently completely lacks a systematic summary of the Outcome Measures used across the studies. A section must be added to categorize and summarize the specific tools used to assess physical and cognitive function.

Reply: Thank you for this comment. We are of the opinion that this manuscript is not about the efficacy of dual-task training, but rather on the methodology. We are worried that, if we deal too deeply with the efficacy of each intervention (which is an added, but not essential, part of the manuscript), it may shift the attention of the reader to other aspects already covered in other articles. Nevertheless, we added a part in the limitation of the study to highlight this issue. We hope the Reviewer understands our decision. The sentence added is: “Outcomes were considered in general, without categorization, to maintain focus on training protocols … and evaluate outcomes more rigorously.”

Discussion

  1. The first paragraph of the discussion should not simply repeat the results but should be a concise synthesis of the findings, pointing out the core characteristics of current DTT research, such as the significant uncertainty and heterogeneity in its protocol design.

Reply: Thank you very much for this comment. We modified the entire first paragraph trying to be more concise. Now it results: “The literature review highlighted heterogeneity and uncertainty in the protocol design. Despite this, from the protocols of the included studies, a SOP is proposed for healthy older adults, it is presented in Table 6 and it is deeply discussed below.”

  1. The discussion needs to compare this review's findings with previous systematic reviews in the field, explicitly stating whether the findings support, supplement, or challenge prior conclusions.

Reply: Thank you very much for this insightful comment. This was a point we intended to address. . Unfortunately, it is challenging to find articles that provide protocols specifically for healthy people in dual-task training, as most focus on populations with disabilities. We included original articles in the manuscript, while reviews on this topic tend to be too general and heterogeneous. Among approximately the 200 reviews available, most include populations with pathology, diverse protocols, and lack of generalizable protocols, making comparison difficult. We appreciate this comment and would be grateful if the Reviewer could recommend any relevant systematic reviews proposing DT training protocols to support or refute our findings. Meanwhile, we have added some reviews to support our findings, where possible, hoping they address the Reviewer’s concern.

  1. This is the section most in need of strengthening. The authors must systematically explore the key issues for constructing an SOP, including task selection, principles of intensity and progression, and training dosage.

Reply: thank you very much. More details have been added in the Discussion to provide explanations for the task selection, principles of intensity and progression, and training dosage. The sentences added and modified have been highlighted in yellow within the Discussion section. Because these additions are distributed throughout the discussion, it is not possible to report them all in this reply. If the Reviewer has further feedback on this point, we would be more than happy to continue working to address this comment. We nevertheless hope the Reviewer appreciates the effort and extensive revisions made.

  1. Based on the foregoing in-depth discussion, the authors should attempt to propose a preliminary, modular SOP framework in the discussion section, demonstrating how the review's findings can be translated into practical guidance.

Reply: Different sentences have been added in the discussion, consistently presenting the manuscript as a practical guidance. We hope the Reviewer can appreciate the effort made to provide such guidance, supported by appropriate literature.

  1. To increase the theoretical depth of the discussion, the authors could delve deeper into the potential neural mechanisms of DTT efficacy, such as neural plasticity and brain network reorganization, and link these theories to the effective training elements observed in the results.

Reply: Thank you very much for this suggestion. We added the following sentences and references in the second part of the discussion. We hope the topic is sufficiently addressed, also considering the length of the discussion: “

Literature shows that DT activity leads to structural brain changes, such as in the gray (visual motion complex area) and white matter volume [150]. Even in older adults, cognitive and exercise training can induce cognitive plasticity [151]. Moreover, older adults demonstrate the ability to increase neural resource allocation [152].

From the findings of this review, DT training also improves static and dynamic postural balance. Literature shows that adding a motor task to a balance task triggers brain network reorganization, increasing interregional connectivity and central resources expansion [153].”

  1. The limitations analysis needs to be more in-depth. In addition to mentioning publication bias, the authors should frankly discuss how the significant heterogeneity among the included studies prevented a quantitative synthesis, thereby limiting the strength and generalizability of the conclusions.

Reply: We added the following sentence in the manuscript to correctly highlight this point: “Narrative synthesis was used, as meta-analysis was not feasible due to heterogeneity in secondary tasks and outcomes measures, limiting the strength of the conclusions. Outcomes were considered in general, without categorization, to maintain focus on training protocols.”

  1. The directions for future research need to be very specific. For example, recommend that “future RCTs should directly compare the effects of different types of cognitive tasks (e.g., spatial memory vs. calculation) when combined with the same motor task,” rather than speaking in generalities.

Reply: As requested, we added this useful indication in the manuscript, thank you. The sentence written in the last part of the discussion is: “Future research should include original field studies and randomized controlled trials directly comparing different types of cognitive tasks (e.g., spatial memory vs. calculation) combined with the same motor tasks.”

Other Details

  1. The entire manuscript has several grammatical errors and typos. It must be thoroughly proofread by a native English speaker or a professional editing service before resubmission.

Reply: Thank you for this comment. The full manuscript was checked and corrected by a professional English speaker. We hope the current form satisfies the Reviewer.

  1. The authors should carefully check the reference list to ensure that every citation is formatted strictly according to the journal's guidelines.

Reply: Thank you. We double-checked the bibliography; we hope that Endnote did not change it again. In the last revision round, we will disable Endnote, so we can check our last version carefully, also from this point of view.

Reviewer 2 Report

Comments and Suggestions for Authors

Totally speaking, the present work is rather significant and meaningful in area of the applied and behavioral psychology, because the authors try to investigate and then to conclude a standard operating procedure for a dual task training to improve older adults' physical and cognitive function by a systematic review. And admittedly, the literature search is comprehensive and standard, the all procedure is rigor and proper, the final results will be reliable and valid, and several necessary and essential findings can be also obtained from this key work. Given those mentioned above, I'm very confident to suggest that the present research is deserved to be accepted and published in this journal. However, a few detailed issues have been discovered in the course of reviewing meanwhile, and thank you!

  1. First and foremost, due to the authors have taken the older adults as the specific research object, so all contents about results and conclusions must be limited in the group of older adults. Otherwise, the present results and conclusions of this research would be inaccurate and invalid, and please pay much attention to this serious problem.
  2. The keywords can be revised as follows: dual task training; physical and cognitive function; older adults; standard operating procedure; systematic review.
  3. Several obvious blank places in text should be effectively addressed.
  4. The title of figure 1. should be correctly placed below this chart.
  5. The table 2 is cross-paged, and the table 3 is not normative and standard.
  6. The second and third paragraph is too long to clearly express relevant contents, so it's necessary to divide into several sections to exhibited.
  7. The cnclusion should be rewritten to better display the mian contents and findings of this work.
  8. The references should be also carefully and seriously checked and rectified by authors, and some detailed errors are existed here.

That's all, and thank you!

Comments on the Quality of English Language

I advise the English language of the present work could be improved to more clearly express the research, and thank you!

Author Response

Totally speaking, the present work is rather significant and meaningful in area of the applied and behavioral psychology, because the authors try to investigate and then to conclude a standard operating procedure for a dual task training to improve older adults' physical and cognitive function by a systematic review. And admittedly, the literature search is comprehensive and standard, the all procedure is rigor and proper, the final results will be reliable and valid, and several necessary and essential findings can be also obtained from this key work. Given those mentioned above, I'm very confident to suggest that the present research is deserved to be accepted and published in this journal. However, a few detailed issues have been discovered in the course of reviewing meanwhile, and thank you!

Reply: Thank you very much for the comment, and for the time and effort spent on our manuscript. It is really appreciated. All changes are highlighted in yellow within the manuscript and a reply is provided below the comment in the reply section.

  1. First and foremost, due to the authors have taken the older adults as the specific research object, so all contents about results and conclusions must be limited in the group of older adults. Otherwise, the present results and conclusions of this research would be inaccurate and invalid, and please pay much attention to this serious problem.

Reply: Thank you very much for this really useful comment. We read and checked all the manuscript to clarify that our SOP is for healthy older adults, we hope the Reviewer appreciates our check. In any case, if we missed some sentences and our findings read as if they are also applicable to other populations, we kindly ask to the Reviewer to highlight them (line number and page) to us, so we can revise accordingly.

  1. The keywords can be revised as follows: dual task training; physical and cognitive function; older adults; standard operating procedure; systematic review.

Reply: Thank you for this comment. To differentiate from the title and make the keywords more comprehensive, we added ‘elderly’ instead of ‘older adults’ and ‘systematic review’. We removed ‘double task’, ‘dual tasking’, and ‘training’.

  1. Several obvious blank places in text should be effectively addressed.

Reply: Thank you for this comment. We reviewed the entire manuscript, trying to avoid as many blank spaces as possible.

  1. The title of figure 1. should be correctly placed below this chart.

Reply: Thank you very much for the comment. We moved the title of Figure 1 below the figure.

  1. The table 2 is cross-paged, and the table 3 is not normative and standard.

Reply: Thank you for this comment, we will double-check the tables in the final editorial stages, if the article is accepted for publication. Thank you for Table 3 (now 5), we corrected it trying to standardize the formatting as recommended.

  1. The second and third paragraph is too long to clearly express relevant contents, so it's necessary to divide into several sections to exhibited.

Reply: Thank you for this comment. The manuscript has been significantly edited according to the other Reviewers’ comments. We modified different sections of the manuscript. We kindly ask to the reviewer if this comment has been solved. Otherwise, we kindly ask to the Reviewer to indicate the page and the line numbers that needs attention. Thank you.

  1. The conclusion should be rewritten to better display the mian contents and findings of this work.

Reply: Thank you very much for this comment, we have made the conclusion section more concise with the relevant contents.

  1. The references should be also carefully and seriously checked and rectified by authors, and some detailed errors are existed here.

Reply: Thank you. We double-checked the bibliography; we hope that Endnote did not change it again. In the last revision round, we will disable Endnote, so we can check our last version carefully, also from this point of view.

That's all, and thank you!

Reviewer 3 Report

Comments and Suggestions for Authors

The authors present a systematic review with the ambitious goal of synthesizing existing dual-task training interventions for healthy older adults and proposing a standard operating procedure (SOP). The topic is very interesting given the increasing focus in interventions for aging populations. The authors also attempt to provide structured guidance for future implementation, which adds extra value. However, the manuscript presents some methodological and reporting gaps that should be addressed. Specifically:

  • The authors state they did not use any risk of bias assessment. According to PRISMA guidelines, even if the purpose is descriptive, quality appraisal is essential to contextualize the findings. It is recommended to include a proper risk of bias assessment or clearly justify why it was not performed and how that affects interpretation.
  • There is no mention or use of GRADE or similar frameworks to assess the certainty of evidence. It is suggested to discuss it as a limitation.
  • There is no protocol registration, I can’t understand why. It is recommended to register the protocol and clarify the process for study selection and data extraction. To my understanding, it is within the journal's policy to accept systematic reviews that are registered.
  • The authors extracted and summarized data narratively but did not provide a clear rationale for grouping or drawing conclusions regarding the SOP. It is suggested to describe the synthesis logic more clearly. Was frequency-based consensus used? Were effect sizes or outcomes considered?
  • The proposed SOP in Table 3 is informative, but its evidence base is not sufficiently explained. For example, why are 12 weeks and two sessions per week recommended? Is this based on mean values? Outcomes? Please openly justify the SOP components using data for example how many studies used each dosage, and with what results? etc Also in the SOP Table it can’t be understood what evidence-based and what expert opinion is. Please clarify.
  • There are a lot of grammatical, syntactic, and lexical issues throughout the text. It is recommended a through language revision from the authors or, if it is not possible, from a professional.
  • The PRISMA checklist has several critical missing elements. Please check it again.
  • In the discussion part, while the ACSM guidelines are briefly mentioned, there is minimal comparison with other relevant international standards or with previous systematic reviews proposing DT protocols. Consider expanding this by comparing the proposed SOP with existing recommendations or highlighting how this work adds clarity or fills gaps in the field.
  • Although limitations are discussed, they could be more critically examined. For example, the lack of risk of bias assessment, absence of effect size reporting, and reliance on narrative synthesis should be emphasized as limitations affecting the strength of conclusions.
  • The discussion does mention the need for more targeted SOPs for specific clinical populations but does not elaborate on how this might be achieved. Propose concrete future research directions, such as standardized trials comparing primary task types, or the role of progression and task difficulty modulation, in order to advance SOP development.

The manuscript makes a promising and important contribution to the scientific literature on cognitive/motor interventions in older adults. However, major revisions are necessary to improve methodological presentation, and overall linguistic quality. I encourage resubmission following these improvements.

Author Response

The authors present a systematic review with the ambitious goal of synthesizing existing dual-task training interventions for healthy older adults and proposing a standard operating procedure (SOP). The topic is very interesting given the increasing focus in interventions for aging populations. The authors also attempt to provide structured guidance for future implementation, which adds extra value. However, the manuscript presents some methodological and reporting gaps that should be addressed.

Reply: Thank you very much for the comment, the time, and effort spent on our manuscript. It is really appreciated. All changes are highlighted in yellow within the manuscript, and a reply is provided below the comment in the reply section.

Specifically:

  • The authors state they did not use any risk of bias assessment. According to PRISMA guidelines, even if the purpose is descriptive, quality appraisal is essential to contextualize the findings. It is recommended to include a proper risk of bias assessment or clearly justify why it was not performed and how that affects interpretation.

Reply: Thank you very much for this comment. It is really useful and appreciated. After this first revision round, because another reviewer highlighted that the methodology is not sufficient for our manuscript to consider it a systematic review, we decided to adopt PRISMA guidelines for a scoping review. Our decision was taken because the manuscript methodology cannot be improved (as the risk of bias assessment and the article quality assessment) and our manuscript perfectly fits as a scoping review. Related to the risk of bias and their quality assessment, the articles included are too heterogeneous, making their evaluation difficult. Different articles’ typology requires different scales, making it difficult to compare the results.

  • There is no mention or use of GRADE or similar frameworks to assess the certainty of evidence. It is suggested to discuss it as a limitation.

Reply: Thank you for this comment. As a scoping review, GRADE or similar frameworks are not required. We hope that the reviewer appreciates our decision to make our manuscript a scoping review. In this way, its rationale will be better justified, making the methodology more appropriate.

  • There is no protocol registration, I can’t understand why. It is recommended to register the protocol and clarify the process for study selection and data extraction. To my understanding, it is within the journal's policy to accept systematic reviews that are registered.

Reply: Thank you again for this comment. The Reviewer is right, the Journal’s policy requires a protocol registration number. As explained to the Journal, it is not our first manuscript attempting to create a protocol and all our previous attempts to register a review of this type on PROSPERO (York) failed. Our work wants to create a protocol and priority is given to studies that want to analyze differences. Consequently, we decided to write our protocol before, but we avoided registering it to PROSPERO. Furthermore, PRISMA for scoping review states that “Indicate whether a review protocol exists”. Following this principle, we clearly stated in the first part of the Materials and Methods that “The work has not been registered on a specific database but the protocol was written before and followed step by step”.

  • The authors extracted and summarized data narratively but did not provide a clear rationale for grouping or drawing conclusions regarding the SOP. It is suggested to describe the synthesis logic more clearly. Was frequency-based consensus used? Were effect sizes or outcomes considered?

Reply: Thank you very much for this comment. It is really appreciated. The SOP was created based on the frequency of the parameters in the included studies. We added a sentence in the methods section to clarify this point: “The SOP was created based on the frequency of the parameters considered (frequency, intensity, time, type, primary and secondary task typology) in the included studies.”

  • The proposed SOP in Table 3 is informative, but its evidence base is not sufficiently explained. For example, why are 12 weeks and two sessions per week recommended? Is this based on mean values? Outcomes? Please openly justify the SOP components using data for example how many studies used each dosage, and with what results? etc Also in the SOP Table it can’t be understood what evidence-based and what expert opinion is. Please clarify.

Reply: Thank you for this comment. We created our SOP based on the frequency of the parameters adopted. Now there are threemore tables that could help the readers to understand how and why we inserted that parameter in our SOP. Furthermore, we added a sentence in the methodology section to facilitate the understanding of how we created the SOP. In any case, the SOP has been also supported with bibliography in the discussion section. We hope the Reviewer can better appreciate our effort to make this process as standard and objective as possible.

  • There are a lot of grammatical, syntactic, and lexical issues throughout the text. It is recommended a through language revision from the authors or, if it is not possible, from a professional.

Reply: Thank you for this comment. The full manuscript was checked and corrected by a professional English speaker. We hope the current form satisfies the Reviewer.

  • The PRISMA checklist has several critical missing elements. Please check it again.

Reply: Thank you very much. We uploaded a new checklist (for scoping reviews).

  • In the discussion part, while the ACSM guidelines are briefly mentioned, there is minimal comparison with other relevant international standards or with previous systematic reviews proposing DT protocols. Consider expanding this by comparing the proposed SOP with existing recommendations or highlighting how this work adds clarity or fills gaps in the field.

Reply: Thank you very much for this useful comment. It is a point that we wanted to answer, also before this comment. Unfortunately, we have difficulty detecting articles that provide protocols for healthy people for a DT training; most of them are for people with disability. Furthermore, the original articles were included in the manuscript, while the reviews on this topic (that could be the only relevant international standards) are too general and heterogeneous. Within about 200 reviews on this topic, most of them include people with pathology, heterogeneous protocols and lack of generalization of the protocols, making it really hard to compare with those studies. We appreciate this comment and if the Reviewer can help us to find some relevant international standards or systematic reviews that propose DT training protocols to support our results, it would be really appreciated. Anyway, as much as possible, we added some reviews to support our findings, hoping that those answer the Reviewer request.

  • Although limitations are discussed, they could be more critically examined. For example, the lack of risk of bias assessment, absence of effect size reporting, and reliance on narrative synthesis should be emphasized as limitations affecting the strength of conclusions.

Reply: Thank you for this comment. We added these limitations in the last part of the discussion. The sentences adopted in the manuscript are: “First, no risk of bias or article quality assessment was performed due to methodological heterogeneity. Second, effect sizes were not reported. The SOP was based only parameter frequency, without considering study sample sizes.”

  • The discussion does mention the need for more targeted SOPs for specific clinical populations but does not elaborate on how this might be achieved. Propose concrete future research directions, such as standardized trials comparing primary task types, or the role of progression and task difficulty modulation, in order to advance SOP development.

Reply: This comment has been really appreciated. With the Reviewer’s permission, we used the concept of this comment and added it in the final section of our discussion. We hope the Reviewer appreciates it. The sentence that we added is: “Standardized trials should better assess the progression, modulation, and task difficulty, tailored to participant characteristics. Only then will true progress in SOP development be achieved.”

The manuscript makes a promising and important contribution to the scientific literature on cognitive/motor interventions in older adults. However, major revisions are necessary to improve methodological presentation, and overall linguistic quality. I encourage resubmission following these improvements.

Reply: Thank you for your trust and belief in our work. We made a great effort to answer all the comments of the Reviewers and the manuscript results have importantly changed (and we hope, improved). We hope the Reviewer can better appreciate this second (revised) version of the manuscript and we want to thank again the Reviewer for their time and insightful comments.

Round 2

Reviewer 3 Report

Comments and Suggestions for Authors

The manuscript has been appropriately reframed and the methodological clarity has improved. Most suggestions including clarification of the SOP rationale, improved reporting structure, addition of limitations, and language editing have been successfully incorporated. I consider that the revised version can be acceptable for publication.